# Molecular Screening of *Echinococcus* spp. and Other Cestodes in Wild Carnivores from Central Italy

**DOI:** 10.3390/vetsci10050318

**Published:** 2023-04-27

**Authors:** Silvia Crotti, Leonardo Brustenga, Deborah Cruciani, Piero Bonelli, Nicoletta D’Avino, Andrea Felici, Benedetto Morandi, Carla Sebastiani, Sara Spina, Marco Gobbi

**Affiliations:** 1Istituto Zooprofilattico Sperimentale dell’Umbria e delle Marche “Togo Rosati” (IZSUM), Via G. Salvemini 1, 06126 Perugia, Italy; s.crotti@izsum.it (S.C.); n.davino@izsum.it (N.D.); a.felici@izsum.it (A.F.); b.morandi@izsum.it (B.M.); c.sebastiani@izsum.it (C.S.); s.spina@izsum.it (S.S.); m.gobbi@izsum.it (M.G.); 2Department of Veterinary Medicine, University of Perugia, Via San Costanzo 6, 06126 Perugia, Italy; leonardo.brustenga@studenti.unipg.it; 3OIE Reference Laboratory for Echinococcosis, Istituto Zooprofilattico Sperimentale della Sardegna “G. Pegreffi”, Via Vienna 2, 07100 Sassari, Italy; piero.bonelli@izs-sardegna.it; 4Centro Nazionale di Referenza per l’Echinococcosi/Idatidosi (Ce.NRE), Istituto Zooprofilattico Sperimentale della Sardegna “G. Pegreffi”, Via Vienna 2, 07100 Sassari, Italy

**Keywords:** echinococcosis, neglected zoonosis, PCR, taeniidae, tapeworms, wild carnivores, wildlife surveillance, one health

## Abstract

**Simple Summary:**

Tapeworm infections are among the most relevant parasitic diseases in both human and animal health. Several tapeworms rely on wild animals to complete their life cycle, among them, taeniids from the Genus *Echinococcus* are particularly important as they are the causative agents of cystic and alveolar echinococcosis. The study’s aim was to perform a molecular screening of fecal samples collected from carcasses of wild carnivores from Central Italy using a multiplex PCR and Sanger sequencing approach. Out of 279 samples, 134 tested positive for either *Taenia* spp. Or *Echinococcus granulosus* sensu lato. Sanger sequencing was performed on every positive sample to produce a taxonomical attribution of the parasitic DNA. *Echinococcus granulosus* sensu stricto (genotype G3) was detected in only one Apennine wolf, whereas no sample tested positive for *E. multilocularis*. Other tapeworms that were commonly found in the sample pool were: *Mesocestoides corti* (syn. *M. vogae*), *M. litteratus*, *Taenia serialis*, and *T. hydatigena*. The results of the survey suggest that *Echinococcus* infections in the study area do not seem to be sustained by sylvatic cycles. The survey corroborates, yet again, the importance of passive surveillance of wild animals that can serve as reservoirs for zoonotic pathogens.

**Abstract:**

Tapeworm infections are among the most relevant parasitic diseases in humans and animals. Tapeworms from the Genus *Echinococcus* are particularly important as they can cause cystic or alveolar echinococcosis. A molecular screening was performed on 279 fecal samples collected from carcasses of wild carnivores from Central Italy using PCR targeting diagnostic fragments of *nad1*, *rrnS*, and *nad5* genes. Samples positive for either *Taenia* spp. or *Echinococcus granulosus* were sequenced to taxonomically identify the parasitic DNA. Of the 279 samples, 134 (48.0%) gave positive results in the multiplex PCR. Only one (0.4%) sample from an Apennine wolf tested positive for *Echinococcus granulosus* sensu stricto (genotype G3), whereas no sample tested positive for *E. multilocularis*. The most frequently detected tapeworms were: *Mesocestoides corti* (syn *M. vogae*) (12.9%), *M. litteratus* (10.8%), *Taenia serialis* (9.3%), and *T. hydatigena* (6.5%), other tapeworms were rarely detected. The results suggest that *Echinococcus* infections in Central Italy do not seem to be sustained by sylvatic cycles, confirming the absence of *E. multilocularis* in Central Italy. The survey corroborates, yet again, the importance of passive surveillance of wild animals that can serve as reservoirs for zoonotic pathogens, especially on wild canids that in other areas are strongly implicated in the transmission of *E. granulosus* and *E. multilocularis*.

## 1. Introduction

Echinococcosis is a parasitic disease sustained by tapeworms belonging to the genus *Echinococcus* (Cestoda, Cyclophyllidaea, Taeniidae). *Echinococcus granulosus* and *Echinococcus multilocularis* are the species of major relevance in human health, causing cystic echinococcosis (CE) and alveolar echinococcosis (AE), respectively [1]. The life cycle of *Echinococcus* is indirect, based on a fecal-oral transmission route and requires the presence of two mammalian hosts: A carnivorous definitive host and an intermediate host, either herbivorous or omnivorous, but generally a prey to the definitive host species. Several species of canids, some felids and hyenas, are reported to be definitive hosts, in which adult worms can be found attached to the small intestine mucosa where they shed gravid proglottids containing eggs, eventually shed together with feces. The oncospheres are contained in resistant eggs that can be accidentally ingested by a wide array of intermediate hosts. In the stomach, the eggs hatch and the activated oncosphere enters the bloodstream allowing for the colonization of viscera in which the metacestode will develop in the form of hydatid cysts. The ingestion of metacestode-infected tissues by definitive hosts closes the cycle with protoscoleces evagination and attachment to the intestine wall [1,2,3]. In the life cycles of *Echinococcus*, humans are accidental intermediate hosts [1]: Clinical outcomes of echinococcosis in humans strictly depend on the parasite species. *Echinococcus granulosus* and *E. multilocularis* are the most widely human-infecting species within the genus [4], but cases of polycystic echinococcosis caused by *Echinococcus vogeli* and *Echinococcus oligarthrus* were reported in Central and Southern America [5], whereas the possible clinical significance of *Echinococcus felidis* and *Echinococcus shiquicus* in humans is still uncertain [3,5]. Overall, clinical trials have shown a mortality rate of 2–4% for CE, that increased relevantly in cases of poor care and inappropriate treatments, whereas 90% of mortality rate after 10–15 years from diagnosis has been shown for AE if left untreated or with limited treatment [4]. For its relevance in human health, echinococcosis has been listed by the World Health Organization (WHO) in a group of 17 neglected zoonoses that should be prioritized for control or elimination by 2050 [6]. The taxonomy of the genus *Echinococcus* has been extensively reviewed in the last decade, especially thanks to molecular biology techniques, yet there is still a wide heterogeneity in the use of terms to define both the parasites and the pathologies they cause [7]. The classification of the paraphyletic taxon *E. granulosus* sensu lato (s.l.) was historically used to identify all the etiological agents responsible for CE onset. In that framework, a classification of *E. granulosus* s.l. in several strains was based on host specificity as follows: G1 (sheep strain), G2 (Tasmanian sheep strain), G3 (buffalo strain), G4 (horse strain), G5 (cattle strain), G6 (camel strain), G7 (pig strain), G8 (American cervid strain), G9 (variant pig or human-pig strain), G10 (Fennoscandian cervid strain) [8,9,10,11,12]. Recently, the study of multiple loci within mitochondrial DNA (mtDNA) and nuclear DNA (nDNA) [3], allowed the subdivision of *E. granulosus* s.l. into: *E. granulosus* sensu stricto (s.s.) (genotypes G1–G3), *E. equinus* (genotype G4), *E. ortleppi* (genotype G5), *E. canadensis* (genotypes G6–G10), and *E. felidis* [13]. Therefore, to the best of the authors’ knowledge, the current taxonomy of the genus *Echinococcus* includes nine species, five of which regroup the 10 previously defined strains, namely: *E. granulosus*; *E. canadensis*; *E. ortleppi*; *E. felidis*; *E. equinus*; *E. multilocularis*; *E. oligarthrus*; *E. vogeli*; *E. shiquicus* [3,7,14].

The main risk factor for echinococcosis infection in humans is close contact with livestock guarding dogs and with dogs that have unsupervised access to the sylvatic environment, which can, in fact, become bridging hosts leading to tapeworm egg shedding in the human environment [15]. Dogs are definitive hosts for both *E. granulosus* s.l., which they contract by eating infected animal organs, and for *E. multilocularis* contracted via the ingestion of small mammals, such as mice and voles [16]. Even though there are reliable estimates on the epidemiology of *Echinococcus* spp. infections within the domestic cycle [5], the extent of the involvement of wild species in the transmission of cystic echinococcosis is still unknown [17]. Aside from the health perspective, parasites are known to also have an impact on the population ecology of host species and *Echinococcus* makes no exception: For example, infection with *E. granulosus* is an important regulator of wolf-moose population dynamics in Quebec, Canada [18].

Along with *Echinococcus*, other tapeworms in the order Cyclophyllidea can be harbored by wild animals that serve as reservoirs. Within Cyclophyllidea, the families that are often found in wildlife are: Taeniidae, Mesocestoididae, and Anoplocephalidae. Taeniidae encompasses four Genera, two of them being *Echinococcus* and the much more speciose *Taenia*, consisting of about 50 species [19] which shares similar life-cycles to the one previously described for *Echinococcus*. Mesocestoididae groups tapeworms that share a three-host life cycle, probably a mite, an herbivore, and eventually a carnivore [20]. Two are the known genera, *Mesocestoides* and *Mesogyna*, but there is almost unanimous consensus over the removal of Mesocestoididae from Cyclophyllidea and the institution of an independent order [21,22]. Anoplocephalidae is a rich family of tapeworms detected in bovids (*Moniezia*), equids (*Anoplocephala* and *Paranoplocephala*), rodents (*Bertiella* and *Diandrya*), but also diffused to other *taxa* like the genus *Atriotaenia* detected in carnivores (e.g., *A. procyonis*, *A. sandgroundi*, *A. incisa*) and bats (*A. hastati*) [23]. Given the epidemiological relevance that many pathogens have in both human and animal health, passive surveillance carried out on carcasses of wild animals has been proposed as a tool able to prevent or, at worst, prepare for possible epidemic outbreaks [24]. The molecular detection of parasites from fecal samples by specific DNA extraction protocol and PCR represents an invaluable asset for epidemiological studies [5]. This is particularly true for wildlife since this diagnostic method would allow for the analysis of animal samples without issues associated with direct interaction with elusive species, and therefore, may allow a greater number of samples to be collected [25,26,27].

Italy is considered an endemic area for cystic echinococcosis [28]. According to both the European Food Safety Authority (EFSA) and the European Center for Disease Control (ECDC), Italy lacks a surveillance system for human CE as well as an epidemiological record of echinococcosis in livestock [29]. Considering the typical domestic life cycle of *E. granulosus*, human infections occur, especially in rural areas characterized by intensive sheep breeding [30]. This scenario is confirmed by looking at the regional infection prevalence throughout Italy, especially in the major islands like Sardinia and Sicily [31,32] where CE is highly endemic. Conversely, the prevalence values from Northern Italy are the lowest at the national level [33]. Extensive research on possible wild definitive hosts of *Echinococcus* in Italy was carried out in the last decades, with a particular focus on the Apennine wolf (*Canis lupus italicus* Altobello, 1921) [34,35,36] and the red fox (*Vulpes vulpes* Linnaeus, 1758) [25,37,38], but no studies seem to have focused on other carnivoran taxa.

Along with the red fox and the Apennine wolf, several other carnivores can be found in Central Italy, some, like the European badger (*Meles meles* Linnaeus, 1758), the stone marten (*Martes foina* Erxleben, 1777), and the pine marten (*Martes martes* Linnaeus, 1758), are common and evenly distributed, whereas others, like the European wildcat (*Felis silvestris silvestris* Schreber, 1777), the European polecat (*Mustela putorius* Linnaeus, 1758), and the least weasel (*Mustela nivalis* Linnaeus, 1766), are less abundant and are much more elusive. Recently, the golden jackal (*Canis aureus* Linnaeus, 1758) has seen a notable range expansion from Northeastern Italy down to Central Italy. While most of these species (e.g., European wildcat, European polecat, least weasel, and pine martens) are found predominantly in natural habitats, the others have adapted to also use urban and suburban areas to hunt or scavenge on human trash. This could be particularly problematic as the increasing contact with livestock or humans can allow the transmission of several pathogens of clinical relevance.

The aim of the study was to carry out a molecular screening using a validated multiplex PCR/Sanger Sequencing approach, able to detect *Echinococcus multilocularis*, *E. granulosus*, and a wide array of other tapeworms, on fecal samples collected from wild carnivores found dead or that died in wildlife rescue centers.

## 2. Materials and Methods

### 2.1. Sample Collection and Conservation

A total of 279 fecal samples were collected from animals either found dead, mainly due to road accidents, or that died in wildlife rescue centers from 2014 to 2023 in Umbria and Marche regions (Central Italy). All animals were subjected to necropsy while still fresh or in very early stages of decomposition, and upon dissection, a fecal sample was extracted from the rectum of each necropsied animal. Animals from wildlife rescue centers died either from traumatic injuries and diseases or were humanely euthanized because the condition they were found in did not allow for rehabilitation and release into nature. No animal was purposefully killed to be enrolled in this study. Carcasses were either found dead and transported by authorized personnel or delivered to the necroscopy facility from the two wildlife rescue centers. Animals that came from wildlife rescue centers and enrolled in the study were not administered any antiparasitic treatment. Environmentally gathered samples were discarded to avoid multiple sampling from the same animal, which could overestimate the frequency of infection in the sample pool. All harvested fecal samples were subsequently subjected to molecular analysis. Sampling consisted of: 135 red foxes (*Vulpes vulpes*), 66 found dead and 69 from wildlife rescue centers; 97 Apennine wolves (*Canis lupus italicus*), 95 found dead and 2 from wildlife rescue centers; 1 golden jackal (*Canis aureus*), found dead; 19 European badgers (*Meles meles*), all from wildlife rescue centers; 11 European wildcats (*Felis silvestris silvestris*), all found dead; 8 pine martens (*Martes martes*), all from wildlife rescue centers; 6 stone martens (*Martes foina*), 3 found dead and 3 from wildlife rescue centers; and 2 European polecats (*Mustela putorius*), all from wildlife rescue centers.

The samples were analyzed for the presence of tapeworm DNA, such as *Taenia* spp., *Echinococcus multilocularis*, and *Echinococcus granulosus* sensu lato (s.l.). Before genomic DNA extraction, each sample was stored in individually labeled sterile zip-lock plastic bags or sterile plastic tubes to prevent contamination and, as a safety precaution, they were stored initially at −80 °C for 10 days [39] and then at −20 °C until DNA extraction.

### 2.2. DNA Extraction and PCR Amplification

DNA was extracted from 0.22 g of feces using the QIAamp DNA fecal Mini Kit (QIAGEN^®^, Hilden, Germany), according to the manufacturer’s instructions and subsequently subjected to multiplex polymerase chain reaction (mPCR) to amplify diagnostic fragments of the mitochondrial genome [40]. Furthermore, a 759 bp fragment of the *nad5* mitochondrial gene, able to identify *E. granulosus* sensu stricto (s.s.) genotypes (G1–G3) [41], was amplified whenever a sample tested positive for *E. granulosus* s.l. Different primer pairs, specific for each target, were used as shown in Table 1.

Multiplex PCR amplifications were carried out on a total volume of 50 μL. The reaction mixture was prepared as follows: 5× Green GoTaq^®^ Flexi Buffer (Promega, Madison, WI, USA), 2 mM of MgCl_2_ (Promega, Madison, WI, USA), 0.2 mM of each dNTP (Global Life Sciences Solutions Operations, Little Chalfont, UK), 8 μL of primer mix (0.3 mM of primers Cest1, Cest2, Cest3, Cest4, and 0.4 mM of Cest5), 1.25 units of GoTaq^®^ G2 Flexi DNA Polymerase (Promega, Madison, WI, USA), 5 μL of template DNA and Ambion^TM^ Nuclease-Free Water (Thermo Fisher Scientific, Austin, TX, USA) to a final volume of 50 μL. All PCR amplifications were performed in a Mastercycler Nexus X2 (Eppendorf AG, Hamburg, Germany) following amplification schemes reported in the references of Table 1. PCR products were run in a 2% agarose gel containing Midori Green Advance (NIPPON Genetics^®^, Europe GmbH, Düren, Germany). Samples that tested positive for *E. granulosus* s.l. were furtherly amplified with a reaction mixture as follows: 5X Green Gotaq^®^ Flexi Buffer (Promega, Madison, WI, USA), 1.5 mM of MgCl_2_ (Promega, Madison, WI, USA), 0.2 mM of each dNTP (Global Life Sciences Solutions Operations, Little Chalfont, UK), 1 mM of each primer (EGnd5F1 and EGnd5R1), 1.5 units of GoTaq^®^ Hot Start DNA Polymerase (Promega, Madison, WI, USA), 2 μL of template DNA and Ambion^TM^ Nuclease-Free Water (Thermo Fisher Scientific, Austin, TX, USA) to a final volume of 50 μL.

### 2.3. Sequencing and Taxonomical Identification

Positive samples detected by PCR were further characterized by Sanger sequencing. PCR reactions were purified using the QIAquick PCR Purification Kit (QIAGEN^®^), according to the manufacturer’s instructions. Quality and quantity of the PCR products were assessed photometrically using a Biophotometer (Eppendorf AG, Hamburg, Germany). Sequencing reactions were carried out in both directions, with the same primers used for PCR amplifications, using BrilliantDye^TM^ Terminator v3.1 Cycle Sequencing Kit (NimaGen^®^, Nijmegen, The Netherlands). Sequencing reactions were run in a 3500 Genetic Analyzer (Applied Biosystem, Foster City, CA, USA). The obtained sequences were analyzed in BioEdit v7.2.5 software [42] and then aligned in the GenBank database [43] using MEGA11 Software v 11.0.13 [44].

## 3. Results

DNA was successfully extracted from all the processed samples since both animals found dead and animals euthanized in wildlife rescue centers were freshly dead, the autolysis processes did not compromise the application of the protocol. Out of 279 fecal samples, 133 (47.7%) were positive for *Taenia* spp. (267 bp) in mPCR, 1 (0.4%) was positive for *Echinococcus granulosus* sensu lato (s.l.) (117 bp), and none for *E. multilocularis*. In detail: 75 of 135 red foxes (55.6%), 45 of 97 Apennine wolves (46.4%), 8 of 11 European wildcats (72.7%), 3 of 19 European badgers (15.8%), and 2 of 6 stone martens (33.3%) were positive for *Taenia* spp., whereas only 1 of 97 Apennine wolf samples (1.0%) was positive for *E. granulosus* s.l. All the pine martens, the European polecats and the golden jackal samples tested negative for all the analyzed loci. To furtherly define *Echinococcus* species, an end-point PCR for *nad5* gene was performed, obtaining a 759 bp amplicon referable to *E. granulosus* sensu stricto (s.s.) (Table 2).

Sequencing analysis of positive samples showed a wide variety of taenids, in particular: *Mesocestoides corti* (syn. *M. vogae*) (*n* = 36, 12.9%), *M. litteratus* (*n* = 30, 10.8%), *T. serialis* (*n* = 26, 9.3%), and *T. hydatigena* (*n* = 18, 6.5%) showed the highest percentages. The only (*n* = 1, 0.4%) *E. granulosus* s.s. sample was identified as genotype G3.

## 4. Discussion

The molecular survey was performed on fecal samples harvested from carcasses of wild carnivores from Central Italy. Even though carcasses were collected in two different ways (animals found dead vs. euthanasia in a wildlife rescue center), all animals enrolled in the study were free-ranging, therefore, there is no reason to believe that parasite detection can be different in that sense. The molecular screening was carried out using an already validated multiplex PCR able to identify *Echinococcus multilocularis*, *E. granulosus* sensu lato (s.l.), and a wide array of parasitic cestodes [40]. Overall, 134 fecal samples tested positive for enteric tapeworms, including *E. granulosus* sensu stricto (s.s.) detected in just one sample from an Apennine wolf, subsequently attributed to genotype G3 by Sanger sequencing. The primers designed to amplify a 267 bp fragment of DNA from *Taenia* spp. tapeworms are also able to amplify DNA of tapeworms from the Genera *Mesocestoides*, *Dipylidium*, and *Diphyllobothrium* [40]. Of 133 samples that showed 267 bp long amplification products referable to *Taenia* spp., only 59 were confirmed by sequencing results to belong to the genus *Taenia*, whereas the remaining samples belonged to the genera *Mesocestoides* (72 samples) and *Atriotaenia* (2 samples).

Even though the analyzed sample is not big enough to perform an adequate perfect test to exclude the presence of tapeworms from the genus *Echinococcus* in the analyzed species, the obtained results corroborate the hypothesis that wild carnivores do not play a substantial role in the life cycle of *Echinococcus* spp. in Central Italy. Moreover, the absence of *E. multilocularis* in the analyzed pool is a comforting result, given the numerous foxes screened. This absence datum is, in fact, in line with the known Italian geographical distribution that reports *E. multilocularis* presence only in Northern Italy [45]. Nonetheless, passive surveillance is of utmost importance to readily identify and manage the emergence of possible infection foci. The detection of an Apennine wolf positive to *E. granulosus* s.s. genotype G3 was not unexpected, in fact, there are reports of Apennine wolves infected with G1 and G3 genotypes in both North Eastern [35] and Central Italy [36,46]. Furthermore, the G3 genotype was also detected in a wild boar (*Sus scrofa* Linnaeus, 1758) from Central Italy [47]. It is reasonable to believe that the infection of *E. granulosus* s.s. (G1–G3) in Apennine wolves can stem from either direct predation on unprotected livestock or from the scavenging of animal remains around sheep (*Ovis aries* Linnaeus, 1758) and goat (*Capra hircus* Linnaeus, 1758) upbringings, but the findings of Di Paolo and colleagues [47] also open the possibility of natural infections of Apennine wolves that acquire the parasite from infected wild ungulates. Wild animals actively search for food near livestock farms, easily finding animal carcasses waiting for disposal or illegally left in the wild to rot. More efficient controls and improved biosecurity measures would, therefore, be beneficial to lower the potential risk of wild animals’ infection by consumption of livestock remains.

Tapeworms from the genus *Mesocestoides* are the most frequent parasitic cestodes of red foxes in Europe [48], and the reason for the massive infection is to be found in their life cycle that needs to exploit a wide array of small vertebrate as second intermediate hosts [49] to develop tetrathyridia inside their coelomic cavity [50]. Among the second intermediate hosts are included: amphibians, reptiles, birds, and small mammals, like rodents, that are all common prey to red foxes. *Taenia polyacantha* develops its larval forms in microtid rodents, such as the bank vole (*Myodes glareolus* Schreber, 1780) [51], whereas *T. pisiformis* uses lagomorphs (rabbits and hares) as intermediate hosts. Both lagomorphs and voles fall within the dietary breadth of the red fox in Italy, therefore, these results were, yet again, not surprising, even though some authors consider red foxes as suboptimal definitive hosts for *T. pisiformis* [52,53]. *Taenia serialis* and *T. ovis*, even though they were previously reported in red foxes [54,55], are uncommon species in the red fox parasitic biocenosis as they are more closely connected with wild and domestic ungulates that are infrequent fox prey. To summarize, tapeworms detected in red foxes are mainly derived from predation on small to medium size vertebrates, such as rodents and lagomorphs, and, to a lesser extent, from scavenging on ungulate carcasses.

Apennine wolves were mostly infected by *T. serialis* and *T. hydatigena*, two species that are frequently detected, respectively, in rodents/lagomorphs and in wild and domestic ungulates [56]. According to Craig and Craig [57] *T. hydatigena* is the most prevalent tapeworm of wolves in the boreal biome. Recent reports of *T. serialis* in an European roe deer (*Capreolus capreolus* Linnaeus, 1758) [58], and in Apennine wolves from Central Italy [59] highlight how even wild ungulates, which are the main prey items for Apennine wolves, can be intermediate hosts for *T. serialis* and, therefore, play a role in wolf infection.

In European wildcats, the most frequently detected tapeworm was *T. taeniformis* which uses rodents as intermediate hosts. This tapeworm was previously reported in European wildcats from both Italy [60] and other European countries like Greece [61], Germany [62], and Croatia [63].

Out of the four species of mustelids tested, tapeworm DNA was detected only in European badgers and stone martens. *Atriotaenia incisa* is a tapeworm already reported in European badgers, it has a life cycle that is still fairly unknown, but it is thought to be using coleopterans as intermediate hosts [64]. The foraging ecology of the badger, which heavily exploits vegetal matter and invertebrates in its diet, can give an explanation for the frequency of detection of *A. incisa* that, in other investigations [64,65,66], is among the predominant parasites. 

Golden jackals are slowly expanding through the Italian peninsula from a natural dispersion phenomenon that started in the Northeastern Italian Alps, probably around 1984 [67]. In the last decade, the golden jackal has significantly dispersed South from the Alps, and recently, two individuals were repeatedly camera-trapped in Tuscany [68]. Even though there are unofficial reports from Latium, that are not yet validated, this finding of a dead golden jackal from Marche region marks the southernmost record of presence, as of March 2023, for the species in the Italian peninsula. A first parasitological assessment of this species in Northern Italy was produced by Beraldo and colleagues [69], but continuative surveillance is of paramount importance as golden jackals from Europe are reported to be definitive hosts for both *Echinococcus multilocularis* [70] and *E. granulosus* [71].

## 5. Conclusions

The results of this survey highlight how carcasses can be used as a cheap and ethical way to obtain fecal samples for parasitic surveillance purposes. Overall, several tapeworms of zoonotic relevance were detected through a molecular approach, giving precious epidemiological data. No *Echinococcus multilocularis* positive sample was detected in the analyzed pool, whereas DNA from only one *E. granulosus* s.s. (G3) positive sample was isolated. It is to be noted that this protocol can be a helpful asset as conventional methods are not always satisfactory to differentiate among tapeworm eggs that are usually undistinguishable in size and morphology. Nevertheless, tapeworm egg shedding can be irregular throughout the life of an infected host, constituting a potential limitation of the protocol. A joint effort of molecular screening and necroscopic inspection is, therefore, advisable to avoid false negatives.

## Figures and Tables

**Table 1 vetsci-10-00318-t001:** Primer sequences of target genes.

Target Species	Target Gene (Amplicon Size)	Primer (5′-3′)	Reference
*E. multilocularis*	*nad1* (395 bp)	Cest1: 5′-TGCTGATTTGTTAAAGTTAGTGATC-3′ Cest2: 5′-CATAAATCAATGGAAACAACAACAAG-3′	[40]
*Taenia* spp.	*rrnS* (267 bp)	Cest3: 5′-YGAYTCTTTTTAGGGGAAGGTGTG-3′ Cest5: 5′-GCGGTGTGTACMTGAGCTAAAC-3′	
*E. granulosus* sensu lato (s.l.)	*rrnS* (117 bp)	Cest4: 5′-GTTTTTGTGTGTTACATTAATAAGGGTG-3′ Cest5: 5′-GCGGTGTGTACMTGAGCTAAAC-3′	
*E. granulosus* sensu stricto (s.s.)	*nad5* (759 bp)	EGnd5F1: 5′-GTTGTTGAAGTTGATTGTTTTGTTTG-3′ EGnd5R1: 5′-GAACACCGGACAAACCAAGAA-3′	[41]

**Table 2 vetsci-10-00318-t002:** Proportion of positive detected samples through a PCR sequencing approach (frequency %, 95% Confidence Interval) by each tapeworm species.

Sequencing Results	Animal Species: Total Samples (Frequency %, 95% Confidence Interval)								Total Positive Samples
	Red Fox 135	Apennine Wolf 97	European Wildcat 11	European Badger 19	Stone Marten 6	Pine Marten 8	European Polecat 2	Golden Jackal 1	
*Atriotaenia* *incisa*	0 (0, 0.0–2.7)	0 (0, 0.0–3.7)	0 (0, 0.0–28.5)	2 (10.5, 1.3–33.1)	0 (0, 0.0–45.9)	0 (0, 0.0–36.9)	0 (0, 0.0–84.2)	0 (0, 0.0–97.5)	2 (0.7, 0.1–2.6)
*Echinococcus* *granulosus* s.s. (G3)	0 (0, 0.0–2.7)	1 (1, <0.1–5.6)	0 (0, 0.0–28.5)	0 (0, 0.0–17.6)	0 (0, 0.0–45.9)	0 (0, 0.0–36.9)	0 (0, 0.0–84.2)	0 (0, 0.0–97.5)	1 (0.4, <0.1–2.0)
*Mesocestoides* *canislagopodis*	1 (0.7, <0.1–4.1)	0 (0, 0.0–3.7)	0 (0, 0.0–28.5)	0 (0, 0.0–17.6)	0 (0, 0.0–45.9)	0 (0, 0.0–36.9)	0 (0, 0.0–84.2)	0 (0, 0.0–97.5)	1 (0.4, <0.1–2.0)
*Mesocestoides corti* (syn. *M. vogae*)	33 (24.4, 17.5–32.6)	1 (1, <0.1–5.6)	2 (18.2, 2.3–51.8)	0 (0, 0.0–17.6)	0 (0, 0.0–45.9)	0 (0, 0.0–36.9)	0 (0, 0.0–84.2)	0 (0, 0.0–97.5)	36 (12.9, 9.2–17.4)
*Mesocestoides* *lineatus*	3 (2.2, 0.4–6.4)	0 (0, 0.0–3.7)	0 (0, 0.0–28.5)	0 (0, 0.0–17.6)	0 (0, 0.0–45.9)	0 (0, 0.0–36.9)	0 (0, 0.0–84.2)	0 (0, 0.0–97.5)	3 (1.1, 0.2–3.1)
*Mesocestoides* *litteratus*	30 (22.2, 15.5–30.2)	0 (0, 0.0–3.7)	0 (0, 0.0–28.5)	0 (0, 0.0–17.6)	0 (0, 0.0–45.9)	0 (0, 0.0–36.9)	0 (0, 0.0–84.2)	0 (0, 0.0–97.5)	30 (10.8, 7.4–15.0)
*Mesocestoides* *melesi*	0 (0, 0.0–2.7)	0 (0, 0.0–3.7)	0 (0, 0.0–28.5)	0 (0, 0.0–17.6)	2 (33.3, 4.3–77.7)	0 (0, 0.0–36.9)	0 (0, 0.0–84.2)	0 (0, 0.0–97.5)	2 (0.7, 0.1–2.6)
*Taenia* *hydatigena*	0 (0, 0.0–2.7)	18 (18.6, 11.4–27.7)	0 (0, 0.0–28.5)	0 (0, 0.0–17.6)	0 (0, 0.0–45.9)	0 (0, 0.0–36.9)	0 (0, 0.0–84.2)	0 (0, 0.0–97.5)	18 (6.5, 3.9–10.0)
*Taenia* *ovis*	3 (2.2, 0.4–6.4)	0 (0, 0.0–3.7)	0 (0, 0.0–28.5)	0 (0, 0.0–17.6)	0 (0, 0.0–45.9)	0 (0, 0.0–36.9)	0 (0, 0.0–84.2)	0 (0, 0.0–97.5)	3 (1.1, 0.2–3.1)
*Taenia* *pisiformis*	2 (1.5, 0.2–5.2)	1 (1, <0.1–5.6)	0 (0, 0.0–28.5)	1 (5.3, 0.1–26.0)	0 (0, 0.0–45.9)	0 (0, 0.0–36.9)	0 (0, 0.0–84.2)	0 (0, 0.0–97.5)	4 (1.4, 0.4–3.6)
*Taenia* *polyacantha*	2 (1.5, 0.2–5.2)	0 (0, 0.0–3.7)	0 (0, 0.0–28.5)	0 (0, 0.0–17.6)	0 (0, 0.0–45.9)	0 (0, 0.0–36.9)	0 (0, 0.0–84.2)	0 (0, 0.0–97.5)	2 (0.7, 0.1–2.6)
*Taenia* *serialis*	1 (0.7, <0.1–4.1)	25 (25.8, 17.4–35.7)	0 (0, 0.0–28.5)	0 (0, 0.0–17.6)	0 (0, 0.0–45.9)	0 (0, 0.0–36.9)	0 (0, 0.0–84.2)	0 (0, 0.0–97.5)	26 (9.3, 6.2–13.4)
*Taenia* *taeniaeformis*	0 (0, 0.0–2.7)	0 (0, 0.0–3.7)	6 (54.5, 23.4–83.3)	0 (0, 0.0–17.6)	0 (0, 0.0-45.9)	0 (0, 0.0–36.9)	0 (0, 0.0–84.2)	0 (0, 0.0–97.5)	6 (2.2, 0.8–4.6)
**Total positive samples**	**75** (55.6, 46.8–64.1)	**46** (47.4, 37.2–57.8)	**8** (72.7, 39.0–94.0)	**3** (15.8, 3.4–39.6)	**2** (33.3, 4.3-77.7)	**0** (0, 0.0–36.9)	**0** (0, 0.0–84.2)	**0** (0, 0.0–97.5)	**134** (48, 42.0–54.1)

## Data Availability

Data is available from the corresponding author upon request.

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
