# Peer review of "Molecular Screening of Echinococcus spp. and Other Cestodes in Wild Carnivores from Central Italy"

_vetsci, 2023, doi:10.3390/vetsci10050318_

Round 1

Reviewer 1 Report

The article demonstrates the use of roadkills as a way of studying the presence of tapeworms in carnivorous wildlife. This is a clever way of obtaining information about animals that are difficult to collect samples from. The aim of the study could be clearer, and this complicates the evaluation of methods and conclusion. The manuscript could be strengthened by a discussion of the strengths and limitations of the convenience sampling.

Simple Summary/Abstract

Is it necessary to mention E. multilocularis if it was not among the findings?

The authors mentions “high prevalence”, but this seems arbitrary. It is suggested to phrase without having to define “high”? F. ex. “…most commonly found in the samples were…”

“infestation” generally refers to ectoparaites. Consider replacing with “infections” throughout the manuscript.

Please decide if you use space between numbers between number and % or not.

Introduction

“…eliminated with the feces.” – if the authors mean that the proglottids are shed in the feces, it is suggested “…shed together with feces.”

“[20].Two” -> “[20]. Two” Please check for lacking spaces (there are more).

The authors write an extensive review on the importance of Echinoccocus as a zoonosis. Since the stated aim is to look into potential alternative sylvian carnivore hosts, it would be nice to have some more background on this. For example, which carnivores besides wolf and red fox are referred to that are considered relevant by the authors? Are any of the carnivores relevant in a One Health context (do they have contact to humans or livestock)? Has the situation changed (new carnivores appeared recently) that make a study more relevant?

Regarding the aim: the aim is currently phrased in a way that makes it fuzzy what is to be achieved in the study. It is suggested to phrase it with a clearer focus on what the primers used could detect. For example, to look for presence/absence of mainly E. multilocularis, E. granulosus and Taenia spp., or to map the genetic diversity tapeworms in species X, Y and Z. This will allow the reader to evaluate if the method used are suitable for the stated purpose and if the conclusions are sound.

Materials and Methods

2.1.

Please define the area of sampling (region, municipality…).

How were the fecal samples taken?

“from 2014 to 2022 were subjected to” -> “from 2014 to 2023. The samples were subjected to…” Then remove the sentence about the golden jackal.

Which of the samples were collected as roadkills and which from a shelter? From what year? This could be a table.

How did the animals in the shelter die (disease, euthanizing (for purpose of study or not), other)?

What state were the dead animals in (fresh, decomposed…)?

How were the animals collected?

2.2.

“Total DNA was extracted” -> “DNA was extracted from X g feces…”

“Eppendorf AG” – please provide the rest of the information.

Results

Was it possible to extract DNA successfully from all samples and do mPCR? If not, please discuss why in the discussion.

In Materials and Methods the authors write 278 stool samples were collected, and 279 stool samples in results. Please explain.

Table 2 show 59 Taenia spp. but the results state 133. Please explain.

Table 2: please provide 95% CI with same number of decimals. Please put 95% CI on samples with 0 positive. Please put 95% CI on Total positive samples. Please correct “Atriotaenia incisa” bold and underlined.

Figure 1: please explain why some data points are shown as host species and others as parasite.

It would be interesting to know whether the parasite findings were attributed to roadkills or shelters. Please add some information about this.

Discussion

Please make the text not bold.

The authors write “Environmentally gathered samples were discarded to avoid multiple sampling of the same individual that could overestimate infestation prevalence in the sample pool.” In the wildlife reserve this makes sense, but could the authors elaborate on the likelihood of sampling the same animal in the wild based on a random fecal sample and the carcass?

Could the authors discuss the importance of the age of the animal when collecting convenience samples for the study and the impact that can have on the results (f. ex. which animals are killed as roadkills and connection between age and encountering a parasite).

Please add some discussion about the sample types (roadkill vs shelter) and what this sample material means for the result obtained.

What is the reason that only half of the samples had sufficient DNA

The authors write “The obtained results clearly show that wild carnivores do not play a substantial role in the life cycle of Echinococcus spp. in Central Italy. Moreover, the complete absence of E. multilocularis in the analyzed pool is a comforting result given the high numerosity of foxes screened and it is in line with the known Italian geographical distribution that reports E. multilocularis presence only in Northern Italy [45].” Have the authors done any calculations of freedom of disease based on the number of samples to support these claims? A perfect test and a 1% prevance in the population would require around 332 randomly collected samples to estimate freedom of disease.

“high numerosity” -> “high number” or “numerous” – the statement is saying the same thing with two words.  

Regarding the introduction of E. granulosus s.s. (G1-G3) in Apennine wolves: is there evidence of practices of slaughtering sheep on farms where meat is left in nature?

Please add some discussion about the results observed in the phylogenetic tree.

Conclusion

The author’s conclusion is a discussion about the use of molecular tools versus morphology and public health importance of Echinococcus. Normally a conclusion would state what can be concluded (proven within reasonable doubt) from the data presented. In this case it can be suggested to conclude that…

…the sample was not able to demonstrate presence or absence of E. multilocularis,

…the sample was sufficient to find E. granulosus s.s. (G3) in wolf,

…that roadkills can be a cheap and ethical way of obtaining samples for surveillance efforts,

…that tapeworms of zoonotic importance can be detected in the samples,

… [method limitations].

Author Response

Response to the Editor and Reviewers

First we wish to thank the Editor and Reviewers for considering our manuscript to be published in Veterinary Sciences and for their precious comments and suggestions.

In this letter we explain, point by point, the details of the revisions to the manuscript and our responses to the referees’ comments.

We also have improved the English language during revision.

Reviewer #1

Simple Summary/Abstract

Is it necessary to mention E. multilocularis if it was not among the findings? 

            We believe that mentioning of E. multilocularis and its clinical relevance is important since the molecular method we have utilized was devised to detect also E. multilocularis. Furthermore, absence data are relevant data themselves, to date there are no reports of E. multilocularis in Central Italy but the pathogen is present in Northeastern Italy.

The authors mentions “high prevalence”, but this seems arbitrary. It is suggested to phrase without having to define “high”? F. ex. “…most commonly found in the samples were…”

            The text was modified as suggested by the reviewer.

“infestation” generally refers to ectoparaites. Consider replacing with “infections” throughout the manuscript.

            The text was modified as suggested by the reviewer.

Please decide if you use space between numbers between number and % or not.

            The space was removed and all the manuscript has been checked to assure there are no spaces between numbers and %.

Introduction

“…eliminated with the feces.” – if the authors mean that the proglottids are shed in the feces, it is suggested “…shed together with feces.”

            The text was modified as suggested by the reviewer.

“[20].Two” -> “[20]. Two” Please check for lacking spaces (there are more).

            Proper spacing has been added where necessary.

The authors write an extensive review on the importance of Echinoccocus as a zoonosis. Since the stated aim is to look into potential alternative sylvian carnivore hosts, it would be nice to have some more background on this. For example, which carnivores besides wolf and red fox are referred to that are considered relevant by the authors? Are any of the carnivores relevant in a One Health context (do they have contact to humans or livestock)? Has the situation changed (new carnivores appeared recently) that make a study more relevant?
            A paragraph better delineating which species of carnivores and how they can play a role in the One Health context was added following the suggestion of the reviewer.

Regarding the aim: the aim is currently phrased in a way that makes it fuzzy what is to be achieved in the study. It is suggested to phrase it with a clearer focus on what the primers used could detect. For example, to look for presence/absence of mainly E. multilocularis, E. granulosus and Taenia spp., or to map the genetic diversity tapeworms in species X, Y and Z. This will allow the reader to evaluate if the method used are suitable for the stated purpose and if the conclusions are sound.
            The aim of the study was rephrased to clarify and allow the reader to better understand the study design.

Materials and Methods

2.1.

Please define the area of sampling (region, municipality…).

            A sentence was added to clarify the sampling area.

How were the fecal samples taken?

            A sentence was added to clarify fecal harvesting methodology.

“from 2014 to 2022 were subjected to” -> “from 2014 to 2023. The samples were subjected to…” Then remove the sentence about the golden jackal.

            The text was corrected following reviewer’s advice.

Which of the samples were collected as roadkills and which from a shelter? From what year? This could be a table.
            The numbers of animals found dead and that died in wildlife rescue centers were added in the text.

How did the animals in the shelter die (disease, euthanizing (for purpose of study or not), other)?

            A sentence was added to clarify that no animal was euthanized for the purpose of this study.

What state were the dead animals in (fresh, decomposed…)?

            A sentence was added to clarify the conservation status of the dead animals.

How were the animals collected?

            A sentence was added to clarify collection methodology.

2.2.

“Total DNA was extracted” -> “DNA was extracted from X g feces…”

            The text was modified as suggested by the reviewer.

“Eppendorf AG” – please provide the rest of the information.
            “Eppendorf AG, Hamburg, Germany” was added where needed, following reviewer’s request.

Results

Was it possible to extract DNA successfully from all samples and do mPCR? If not, please discuss why in the discussion.

            A sentence was added to clarify that DNA was successfully extracted from all the processed samples.

In Materials and Methods the authors write 278 stool samples were collected, and 279 stool samples in results. Please explain.

            There was a typing error in the Materials and Methods section, the total number of samples is indeed 279. Material and Methods section was corrected.

Table 2 show 59 Taenia spp. but the results state 133. Please explain.

            Table 2 shows the results of Sanger sequencing performed to provide a reliable taxonomical attribution of the positive samples. It is noted by the original authors of the mPCR protocol (Trachsel et al. 2007, reference n. 40) that the primers designed to amplify Taenia spp. DNA are also able to amplify DNA of tapeworms from the Genera Mesocestoides, Dipylidium and Diphyllobothrium. A total of 133 samples produced a 267 bp long product referable to Taenia sp. according to Trachsel protocol but, upon sequencing, only 59 of them were identified as tapeworms from the genus Taenia. The remaining samples belonged to the Genera Mesocestoides (72 samples) and Atriotaenia (2 samples). To better clarify our findings this aspect was furtherly explained in the Discussion section.

Table 2: please provide 95% CI with same number of decimals. Please put 95% CI on samples with 0 positive. Please put 95% CI on Total positive samples. Please correct “Atriotaenia incisa” bold and underlined.
             “Atriotaenia incisa” was given the correct formatting. Confidence Intervals were removed from the Table 2, and throughout the manuscript the proportion of positive samples were considered as frequency of tapeworms occurring in our study population and not as prevalence due to the type of this secondary-data retrospective study. 

Figure 1: please explain why some data points are shown as host species and others as parasite.
            The data points that were not highlighted represent GenBank entries (along with their code), highlighted and color coded were the tapeworm sequences found in carnivoran hosts. We produced the phylogenetic tree to make sense of the phylogenetic relations among the species detected in our study. Since no phylogenetic analysis was performed in the samples obtained, we decided to completely remove Figure 1 from the manuscript as we concur it was not very informative.

It would be interesting to know whether the parasite findings were attributed to roadkills or shelters. Please add some information about this.
            Both animals from wildlife rescue centers and animals found already dead have the same probability of harboring the parasites since the animals that were rescued from the wildlife rescue centers and were enrolled in this study did not live long enough to be treated for possible tapeworm infections. 

Discussion 

Please make the text not bold.

            The text was formatted in not bold font.

The authors write “Environmentally gathered samples were discarded to avoid multiple sampling of the same individual that could overestimate infestation prevalence in the sample pool.” In the wildlife reserve this makes sense, but could the authors elaborate on the likelihood of sampling the same animal in the wild based on a random fecal sample and the carcass?
            We decided to limit the sampling pool to fecal sample extracted from necropsies to avoid multiple sampling of the same individual. We believe that estimating the likelihood of sampling of the same individual in enclosed wildlife reserves and in the wild environment is difficult to do, especially without a microsatellite typing or trail cam identification support. This was out of the scope and possibilities of our study, therefore we decided to stick to sampling carcasses.

Could the authors discuss the importance of the age of the animal when collecting convenience samples for the study and the impact that can have on the results (f. ex. which animals are killed as roadkills and connection between age and encountering a parasite).
            Unfortunately we do not have a hold of the requested data. The paper stems from passive surveillance data therefore some information was not registered at the time of sampling.

Please add some discussion about the sample types (roadkill vs shelter) and what this sample material means for the result obtained.
            A paragraph on strengths and limitations of the protocol and on the sampling, method was added in the Conclusion section. As highlighted in a prior answer to your comment we don’t believe that a difference can be made out of animals found dead and animals dead in wildlife rescue centers.

What is the reason that only half of the samples had sufficient DNA

            As added in the results, DNA was extracted from all the samples processed but only 134 tested positive for tapeworm DNA. This does not mean that sufficient DNA was extracted from half of the samples, but that only about half of the samples tested positive for tapeworm DNA using Trachsel’s mPCR protocol. 

The authors write “The obtained results clearly show that wild carnivores do not play a substantial role in the life cycle of Echinococcus spp. in Central Italy. Moreover, the complete absence of E. multilocularis in the analyzed pool is a comforting result given the high numerosity of foxes screened and it is in line with the known Italian geographical distribution that reports E. multilocularis presence only in Northern Italy [45].” Have the authors done any calculations of freedom of disease based on the number of samples to support these claims? A perfect test and a 1% prevance in the population would require around 332 randomly collected samples to estimate freedom of disease. 
            Since no calculation of freedom of disease was carried out, the paragraph was rephrased. Furthermore, there is a large body of evidence (and our results align with what previously published by other authors) corroborating the fact that Central Italy is an area unaffected by Echinococus multilocularis.

“high numerosity” -> “high number” or “numerous” – the statement is saying the same thing with two words.

            The text was modified following reviewer’s suggestion.  

Regarding the introduction of E. granulosus s.s. (G1-G3) in Apennine wolves: is there evidence of practices of slaughtering sheep on farms where meat is left in nature?

On small farms in Central Italy, dead animals do not always comply with the EU regulation on animal by-products. There are no official reports on this phenomenon, but dead animals are often abandoned in the pasture or left for days near the farm, eaten by farm dogs and wildlife.

Please add some discussion about the results observed in the phylogenetic tree.
            We did not carry out a phylogenetic analysis to assess the genetic diversity of the detected parasites at sample level. Therefore, we decided to completely remove Figure 1 from the manuscript as we concur it was not very informative.

Conclusion

The author’s conclusion is a discussion about the use of molecular tools versus morphology and public health importance of Echinococcus. Normally a conclusion would state what can be concluded (proven within reasonable doubt) from the data presented. In this case it can be suggested to conclude that…

…the sample was not able to demonstrate presence or absence of E. multilocularis,

…the sample was sufficient to find E. granulosus s.s. (G3) in wolf,

…that roadkills can be a cheap and ethical way of obtaining samples for surveillance efforts,

…that tapeworms of zoonotic importance can be detected in the samples,

… [method limitations].
            The Conclusions have been updated as requested by the reviewer.

Reviewer 2 Report

March 2023

I reviewed manuscript “Molecular screening of Echinococcus spp. and other cestodes in wild carnivores from Central Italy” by Crotti et al. Overall, this paper is nicely written, flows well, and provides pertinent information about tapeworm infections in wild definitive hosts in Italy, which is considered an endemic area for cystic echinococcosis. The results provide additional information about the presence, prevalence, and persistence of these tapeworms across the globe.

I have a few technical comments. It was a bit challenging to review a paper without line numbers, but that appears to be a choice by the journal. I greatly prefer the standard convention of the Oxford comma in lists, which is not included in this paper – that is ultimately up the authors or the journal. There were several sentences where the phrasing is slightly awkward and too wordy; I make suggestions to revise some of these sentences, but not all.

Below are specific comments to help improve the manuscript.

SUMMARY

·      Line “The results of the survey are comforting and highlight that Echinococcus infestations in the study area do not seem to be sustained by sylvatic cycles.” Revise to “The results of the survey suggest that Echinococcus infestations in the study area do not seem to be sustained by sylvatic cycles.”

ABSTRACT

·      Line “The tapeworms detected with high prevalence were: …” revise to “The tapeworms detected with higher prevalence were: …”. I don’t necessarily consider 6.5% to be high prevalence, but it is relatively higher than the prevalence of Echinococcus spp in your sample.

·      Same as in the Summary, I suggest removing that the “results are comforting” (my personal preference, but ultimately up to you).

INTRODUCTION

·      Line “Clinical outcomes of echinococcosis in humans are strictly depending on the parasite species” revise to “Clinical outcomes of echinococcosis in humans strictly depends on the parasite species”

·      Line “For its relevance in human health, echinococcosis has been listed by the World Health Organization (WHO) in a group of 17 neglected zoonoses that will need to be either controlled or eliminated by 2050 [6].” I am not sure what is meant by “need to be” here – either describe why they “need” to be controlled or eliminated, or change “need to be” to something like “should be prioritized for control or elimination…”

·      I do not believe the word “dishomogeneity” is proper – please use “heterogeneity” or something else

·      Line “The main risk factor in the spreading of echinococcosis is the close contact with livestock guarding dogs and with dogs that have unsupervised access to the sylvatic environment, that can in fact become bridging hosts leading to tapeworm egg shedding in the human environment and can cause human infections [15].” Please specify that you are referring to the risk of humans acquiring echinococcosis here.

·      The paragraph starting with line “Along with Echinococcus, other tapeworms in the order Cyclophyllidea are harbored by wild animals that serve as reservoirs to sustain sylvatic parasitic cycles.” is a little clunky and could use revision. For example, please add a comma in this line: “Taeniidae encompasses four Genera, two of them being…”; and revise “Mesocestoididae groups tapeworms sharing a three-host life-cycle…” to “Mesocestoididae group tapeworms share a three-host life-cycle…”.

·      Line “This is particularly true for wildlife, since this diagnostic method would allow the analysis of a large number of animals excluding sampling issues associated with direct interaction with elusive species [25–27].” Please revise to something like: “This is particularly true for wildlife, since this diagnostic method would allow for the analysis of animal samples without issues associated with direct interaction with elusive species, and therefore may allow a greater number of samples to be collected [25–27].”

·      Line “…especially in the major islands i.e. Sardinia and Sicily…” revise to “…especially in the major islands like Sardinia and Sicily…”

·      Please list the carnivore species that live in Central Italy in the last paragraph of the Introduction. This sentence could also be revised and broken into two sentences to improve the flow.

METHODS

·       Please use “fecal” instead of “stool” to remain consistent.

·       Why not look for the worms themselves in the GI tract instead of using fecal samples? Although the methods are sound for tapeworm detection and identification from the fecal samples, the animals are already dead and could be necropsied. CoproDNA is relatively low quality, and egg shedding can be irregular, therefore examining the GI tracts may have produced better results (i.e., fewer false-negatives).

·       It is important to describe how samples were collected from carcasses. Please include details about this.

·       Why did you perform the phylogenic analysis? Please add a rationale somewhere in the Introduction or Methods.

RESULTS

·       Please use “fecal” instead of “stool” to remain consistent.

·       Please specify that each fecal sample represented ONE individual. It should be clear whether prevalence is true prevalence or fecal prevalence.

·       Figure 1 is not particularly useful. The text is far too small to read. Why not include other outgroup samples here? I suggest revising this and potentially including outgroups that make sense for these parasites.

·       Similar to my comment on Figure 1, I am not sure what implications the phylogenetic work here has. How does this advance our knowledge about these parasites? A strong rationale and statements about results and implications should be presented.

DISCUSSION

·       The first paragraph is bolded font – not sure why

·       The first paragraph is very repetitive from other parts of the text. I suggest reducing this repetition and highlighting the most important parts of the study here.

·       In the first paragraph, this line is suited for the Methods, not Discussion: “Environmentally gathered samples were discarded to avoid multiple sampling of the same individual that could overestimate infestation prevalence in the sample pool.”

·       Please use “infected” instead of “infested” to kept terminology consistent.

·       Again, why did you perform the phylogenetic analysis? It is not talked about in the Discussion so it is not clear why it was important here or how it relates to other literature.

Author Response

Response to the Editor and Reviewers

First we wish to thank the Editor and Reviewers for considering our manuscript to be published in Veterinary Sciences and for their precious comments and suggestions.

In this letter we explain, point by point, the details of the revisions to the manuscript and our responses to the referees’ comments.

We also have improved the English language during revision.

Reviewer #2

SUMMARY

  • Line “The results of the survey are comforting and highlight that Echinococcus infestations in the study area do not seem to be sustained by sylvatic cycles.” Revise to “The results of the survey suggest that Echinococcus infestations in the study area do not seem to be sustained by sylvatic cycles.”
    The text was modified as suggested by the reviewer.

ABSTRACT

  • Line “The tapeworms detected with high prevalence were: …” revise to “The tapeworms detected with higher prevalence were: …”. I don’t necessarily consider 6.5% to be high prevalence, but it is relatively higher than the prevalence of Echinococcus sppin your sample.
                The text was modified as suggested by the reviewer.
  • Same as in the Summary, I suggest removing that the “results are comforting” (my personal preference, but ultimately up to you).
    The text was modified as suggested by the reviewer.

INTRODUCTION

  • Line “Clinical outcomes of echinococcosis in humans are strictly depending on the parasite species” revise to “Clinical outcomes of echinococcosis in humans strictly depends on the parasite species”
    The text was modified as suggested by the reviewer.
  • Line “For its relevance in human health, echinococcosis has been listed by the World Health Organization (WHO) in a group of 17 neglected zoonoses that will need to be either controlled or eliminated by 2050 [6].” I am not sure what is meant by “need to be” here – either describe why they “need” to be controlled or eliminated, or change “need to be” to something like “should be prioritized for control or elimination…”
    The text was modified following reviewer’s suggestion.
  • I do not believe the word “dishomogeneity” is proper – please use “heterogeneity” or something else
    The text was modified as suggested by the reviewer.
  • Line “The main risk factor in the spreading of echinococcosis is the close contact with livestock guarding dogs and with dogs that have unsupervised access to the sylvatic environment, that can in fact become bridging hosts leading to tapeworm egg shedding in the human environment and can cause human infections [15].” Please specify that you are referring to the risk of humans acquiring echinococcosis here.
    The sentence was rephrased to better clarify the concept.
  • The paragraph starting with line “Along with Echinococcus, other tapeworms in the order Cyclophyllidea are harbored by wild animals that serve as reservoirs to sustain sylvatic parasitic cycles.” is a little clunky and could use revision. For example, please add a comma in this line: “Taeniidae encompasses four Genera, two of them being…”; and revise “Mesocestoididae groups tapeworms sharing a three-host life-cycle…” to “Mesocestoididae group tapeworms share a three-host life-cycle…”.
    The paragraph was slightly rephrased to better expose the concepts.
  • Line “This is particularly true for wildlife, since this diagnostic method would allow the analysis of a large number of animals excluding sampling issues associated with direct interaction with elusive species [25–27].” Please revise to something like: “This is particularly true for wildlife, since this diagnostic method would allow for the analysis of animal samples without issues associated with direct interaction with elusive species, and therefore may allow a greater number of samples to be collected [25–27].”
    The paragraph was rephrased as suggested by the reviewer.
  • Line “…especially in the major islands i.e. Sardinia and Sicily…” revise to “…especially in the major islands like Sardinia and Sicily…”
    The sentence was modified as suggested by the reviewer.
  • Please list the carnivore species that live in Central Italy in the last paragraph of the Introduction. This sentence could also be revised and broken into two sentences to improve the flow.
    A paragraph better delineating which species of carnivores and how they can play a role in the One Health context was added following the suggestion of the reviewer.

METHODS

  • Please use “fecal” instead of “stool” to remain consistent.
    The text was modified as requested by the reviewer.
  • Why not look for the worms themselves in the GI tract instead of using fecal samples? Although the methods are sound for tapeworm detection and identification from the fecal samples, the animals are already dead and could be necropsied. CoproDNA is relatively low quality, and egg shedding can be irregular, therefore examining the GI tracts may have produced better results (i.e., fewer false-negatives).

We agree with your point of view, but our institution works on surveillance plans for both domestic and wild animals. Inspecting the GI tracts of all the animals searching for adult forms of parasites and then performing morphological and molecular analyses is a time-consuming effort. As we added in the Conclusions section, we are aware that our method could be less sensitive than an integrated approach of morphology and molecular analysis. We still decided to opt for a time-efficient molecular approach to have a general idea of the common tapeworms circulating within carnivores in Central Italy. It will be our responsibility to put efforts in a project dealing with a more sensitive integrated approach.

  • It is important to describe how samples were collected from carcasses. Please include details about this.
    Details about sample collection were added in the Material and Method section.
  • Why did you perform the phylogenic analysis? Please add a rationale somewhere in the Introduction or Methods.

We did not carry out a phylogenetic analysis to assess the genetic diversity of the detected parasites at sample level, but to give a visual information of the relations among the parasites detected. We concur that our phylogenetic tree could not be very informative therefore, we decided to completely remove Figure 1 from the manuscript, adapting the text.

RESULTS

  • Please use “fecal” instead of “stool” to remain consistent.
    The text was modified as requested by the reviewer.
  • Please specify that each fecal sample represented ONE individual. It should be clear whether prevalence is true prevalence or fecal prevalence.
    The fact that each fecal sample represents one individual was specified along with the details of sample collection in the Material and Methods section.
  • Figure 1 is not particularly useful. The text is far too small to read. Why not include other outgroup samples here? I suggest revising this and potentially including outgroups that make sense for these parasites.
    As reported in prior responses to the comments above, we concur that our phylogenetic tree could not be very informative therefore, we decided to completely remove Figure 1 from the manuscript, adapting the text.
  • Similar to my comment on Figure 1, I am not sure what implications the phylogenetic work here has. How does this advance our knowledge about these parasites? A strong rationale and statements about results and implications should be presented.
    We produced the phylogenetic tree to make sense of the phylogenetic relations among the species detected in our study. Since no phylogenetic analysis was performed in the samples obtained and the phylogenetic tree did not advance the knowledge about the detected parasites, we decided to completely remove Figure 1 from the manuscript as we concur it was not very informative.

DISCUSSION

  • The first paragraph is bolded font – not sure why
    The text was formatted in not bold font.
  • The first paragraph is very repetitive from other parts of the text. I suggest reducing this repetition and highlighting the most important parts of the study here.
    The first paragraph of discussions was decluttered as requested by the reviewer.
  • In the first paragraph, this line is suited for the Methods, not Discussion: “Environmentally gathered samples were discarded to avoid multiple sampling of the same individual that could overestimate infestation prevalence in the sample pool.”
    The sentence was moved to the Materials and Methods section.
  • Please use “infected” instead of “infested” to kept terminology consistent.
    The text was modified as requested by the reviewer.
  • Again, why did you perform the phylogenetic analysis? It is not talked about in the Discussion so it is not clear why it was important here or how it relates to other literature.
    We produced the phylogenetic tree to make sense of the phylogenetic relations among the species detected in our study. Since no phylogenetic analysis was performed in the samples obtained and the phylogenetic tree did not advance the knowledge about the detected parasites, we decided to completely remove Figure 1 from the manuscript as we concur it was not very informative.

Round 2

Reviewer 1 Report

The authors have improved the manuscript in many places and made it more transparent. However, the manuscript would be even stronger if uncertanties were shown, and allowing the reader to review the relevance of sampling sources.

Materials and Methods

2.1.

Suggestion: “…a fecal sample, one for each necropsied animal, was extracted from the rectum.” -> “…a fecal sample was extracted from the rectum of each necropsied animal.”

“in Nature” -> “into nature”

“this study, carcasses” -> “this study. Carcasses”

“multiple sampling of the same individual” -> “multiple sampling from the same animal”

“(Canis lupus italicus), 95 found” -> “(Canis lupus italicus), 95 found”

Results

Table 2: The uncertainty of the findings is quite important to display, especially for the negatives (0%). The reason for this is that the authors look for rare occurrences (E. multilocularis) in a limited number of convenience samples, and present a wish to determine something about the presence/absence of Eccinoccocus. The large 95% CI from a limited number of samples will show the reader the confidence this data can be used to state anything regarding presence/absence in the examined hosts. It is therefore recommended to add the 95% CI for all frequencies, including zeros.

Regarding informing the reader which results came from roadkills or shelters: the comment was not necessarily directed towards a potential anthelmintic treatment, but rather that they come from different sources/environment (even if those animals in shelters previously have been in the wild). If the results show a difference in the two sampling sources it would be important to reflect on this (f. ex. did decomposing carcasses found in the wild versus freshly dead animals in the shelter affect the results). If the authors think there is no difference between the sources, please state so in the discussion, but please give the reader the possibility to make their own interpretation. The reference to having two sampling sources was not clear in the strengths and limitations in the conclusion. Please clarify in the results, discussion and conclusion.

Discussion

Regarding absence of E. multilocularis: As previously stated, the authors have had access to a limited number of samples, because they are hard to obtain systematically. This is fair, considering the samples are limited. However, this makes the error (95% CI) even more important to list in table 2. Please consider the sentence “The obtained results suggest that wild carnivores do not play a substantial role in the life cycle of Echinococcus spp. in Central Italy.” in the context of needing a sample size of >300 animals with a perfect test to likely find 1 positive animal with E. multilocularis. The sample is not big enough to exclude presence in the population, but it can support that the infection it is rare in the bigger samples (such as fox, wolf). Please think how to phrase this.

The information about animals left to be consumed by wildlife is an important practice in the epidemiology of the parasites (it is a potential intervention). Please include this.

Author Response

Response to Reviewer #1

 Thank you for your additional comments.

 All your requests were addressed in the text and all authors approved them.
